# Diagnostic Image Quality of a Low-Field (0.55T) Knee MRI Protocol Using Deep Learning Image Reconstruction Compared with a Standard (1.5T) Knee MRI Protocol

**DOI:** 10.3390/jcm12051916

**Published:** 2023-02-28

**Authors:** Ingo Lopez Schmidt, Nina Haag, Iram Shahzadi, Lynn Johann Frohwein, Claus Schneider, Julius Henning Niehoff, Jan Robert Kroeger, Jan Borggrefe, Christoph Moenninghoff

**Affiliations:** 1Department of Radiology, Neuroradiology and Nuclear Medicine, Johannes Wesling University Hospital, Ruhr University Bochum, 32429 Minden, Germany; 2Siemens Healthcare, GmbH, 91052 Erlangen, Germany

**Keywords:** magnetic resonance imaging, deep learning reconstruction, low-field MRI, knee, diagnostic imaging

## Abstract

Objectives: Low-field MRI at 0.55 Tesla (T) with deep learning image reconstruction has recently become commercially available. The objective of this study was to evaluate the image quality and diagnostic reliability of knee MRI performed at 0.55T compared with 1.5T. Methods: A total of 20 volunteers (9 female, 11 male; mean age = 42 years) underwent knee MRI on a 0.55T system (MAGNETOM Free.Max, Siemens Healthcare, Erlangen, Germany; 12-channel Contour M Coil) and a 1.5T scanner (MAGNETOM Sola, Siemens Healthcare, Erlangen, Germany; 18-channel transmit/receive knee coil). Standard two-dimensional (2D) turbo spin echo (TSE), fat-suppressed (fs) proton density-weighted (PDw), T1w TSE, and T2w TSE sequences were acquired in approximately 15 min. In total, 2 radiologists blinded to the field strength subjectively assessed all MRI sequences (overall image quality, image noise, and diagnostic quality) using a 5-point Likert scale (1–5; 5 = best). Additionally, both radiologists evaluated the possible pathologies of menisci, ligaments, and cartilage. Contrast ratios (CRs) of different tissues (bone, cartilage, and menisci) were determined on coronal PDw fs TSE images. The statistical analysis included Cohen’s kappa and the Wilcoxon rank sum test. Results: The overall image quality of the 0.55T T2w, T1w, and PDw fs TSE sequences was diagnostic and rated similar for T1w (*p* > 0.05), but lower for PDw fs TSE and T2w TSE compared with 1.5T (*p* < 0.05). The diagnostic accordance of meniscal and cartilage pathologies at 0.55T was similar to 1.5T. The CRs of the tissues were not significantly different between 1.5T and 0.55T (*p* > 0.05). The inter-observer agreement of the subjective image quality was generally fair between both readers and almost perfect for the pathologies. Conclusions: Deep learning-reconstructed TSE imaging at 0.55T yielded diagnostic image quality for knee MRI compared with standard 1.5T MRI. The diagnostic performance of meniscal and cartilage pathologies was equal for 0.55T and 1.5T without a significant loss of diagnostic information.

## 1. Introduction

Magnetic resonance imaging (MRI) is the most widely used non-invasive imaging technique for assessing intra-articular injuries of the knee [1]. Meniscal, ligamentous, and cartilage disorders can be clearly visualized due to the excellent tissue contrast and the high spatial resolution [2,3].

Clinical MR imaging is currently dominated by 1.5T and 3.0T whole-body scanners. The increase in field strength leads to a higher signal-to-noise ratio (SNR), thus resulting in a higher achievable spatial resolution. Thereby, the visualization and depiction of the details of ligaments, tendons, and cartilage in musculoskeletal imaging are improved [4]. For imaging and diagnostic evaluations of knee joint cartilage, the superiority of 3.0T over 1.5T MRI has already been documented [5,6].

As a counterpoint to the development of MRI systems of increasing magnetic field strengths, low-field MRI is currently undergoing a renaissance due to image optimization with the aid of artificial intelligence as well as improvements in field homogeneity [7,8]. In addition to hardware factors such as more powerful magnets, coil systems, and lower helium consumption, there are new image reconstruction techniques based on deep learning algorithms that are expected to make the performance of low-field MRI scanners more powerful and cost-effective. The latter has the potential to tackle three limiting factors of MR imaging simultaneously: the image resolution, SNR, and acquisition speed [9]. These improvements have led to a substantial increase in the SNR at low-field, allowing images to be acquired with an excellent diagnostic quality in a reasonable scan time [4].

In addition, low-field MRI systems offer several advantages compared with high-field systems. Due to the proportionality between the field strength and magnetic susceptibility, susceptibility artifacts are reduced at low-field. This aspect especially concerns the visualization of tissue near metal implants and at air–tissue interfaces. Promising initial results were obtained for lung imaging at 0.55T [10,11]. In addition, the possible use of a larger bore size of up to 80 cm can increase the potential for MRI-guided interventional procedures at low-field [12]. A combined improvement in patient comfort, acceptance, and safety is also postulated by the manufacturers. Further advantages of low-field systems are lower costs and logistical requirements for installation, setup, operation, and maintenance; reduced-siting logistics offer the possibility of easier access to MRI worldwide [4,9].

Over the last three decades, several studies have been published indicating the high diagnostic performance of low-field MRI systems for the musculoskeletal system [13,14]. For specific knee examinations, the reliability of low-field MRI examinations has been demonstrated with regard to meniscal and ligamentous lesions in prototype scanners [15]. However, there were marked limitations in the detection of cartilage lesions, as also shown by Lee et al. [16]. Due to new developments such as improved flexible coils and deep learning-based reconstruction algorithms, the validity of older comparative studies is limited [4].

In the present prospective study, we compared MRI scans of the knee joint acquired with a clinical low-field (0.55T) MRI scanner using deep learning reconstruction and a standard (1.5T) MRI scanner. We performed a thorough subjective and objective image analysis at both field strengths, including the image quality of the MRI sequences and contrast ratios of various tissues, and compared the diagnostic accordance of anatomic structures and pathological changes of the knee joint.

## 2. Materials and Methods

### 2.1. Subjects and MR Imaging

The study was conducted according to the guidelines of the Declaration of Helsinki and approved by the institutional review board. Examinations were performed after receiving written confirmed consent from all volunteers.

We examined both knees of 20 study participants (9 female, 11 male; mean age of 42) without acute knee pain or acute injury. MRI studies were obtained using 0.55 and 1.5T whole-body scanners (MAGNETOM Free.Max and MAGNETOM Sola, Siemens Healthcare, Erlangen, Germany) in a randomized order. The 0.55T MRI scanner was equipped with a flexible 12-channel contour coil and the 1.5T MRI scanner with an 18-channel phased array transmit/receive knee coil (Siemens Healthcare, Erlangen, Germany). We used the manufacturer’s predefined scan parameters to produce clinically useful examination protocols at both field strengths in a comparable examination time. The MRI protocol consisted of three different turbo spin echo (TSE) sequences—namely, proton density-weighted (PDw) fat-suppressed (fs) sequences in a coronal, transversal, and sagittal orientation—supplemented by a coronal T1w and a sagittal T2w sequence. The scans at 0.55T were acquired and reconstructed using a deep learning pipeline (Deep Resolve Gain and Sharp, Siemens Healthcare, Erlangen, Germany). Deep resolve gain, applied at 0.55T in this study, incorporates specific noise maps that are acquired at the same time as the original raw data directly into the image reconstruction. Consequently, the reconstruction algorithm leads to a more homogenous de-noising whilst taking local noise variations into account [17,18]. Furthermore, simultaneous multislice imaging (SMS) was used at 0.55T to excite and acquire several slices at the same time to speed up the imaging. At both field strengths, the data were reconstructed using the slice generalized autocalibrating partial parallel acquisition (GRAPPA) technique for parallel imaging to increase the spatial resolution in the same examination time [19]. All MRI sequences at 0.55T were obtained with a uniform voxel size of 0.3 × 0.3 × 3.0 mm^3^ whereas the in-plane resolution of the 1.5T sequences was minimally larger, between 0.4 × 0.4 × 3.0 mm^3^ and 0.5 × 0.5 × 3.0 mm^3^.

The detailed sequence parameters are shown in Table 1. 

### 2.2. Qualitative MR Image Analysis

In total, 2 experienced board-certified radiologists (reader 1 with 10 years and reader 2 with 12 years of experience of musculoskeletal MRI readings) evaluated all acquired MR images on a syngo.via workstation (Siemens Healthcare, Erlangen, Germany). The MR scans obtained at 0.55T and 1.5T were evaluated by both readers in a random order regarding the field strengths, study participants, and side of the examined knee joint. The radiologists were blinded to the personal data of the study participants, to the field strength, and to the sequence parameters. 

The radiologists assessed all sequences of each MRI study regarding the image quality and visualization of the anatomic structures. The presence of pathological disorders of the menisci, ligaments, and cartilage was also evaluated by both readers. 

The rating of the image quality included the following criteria: the extent of the image noise; the contrast between the cartilage and synovial fluid; edge sharpness for sagittal PDw fs, coronal T1w, and sagittal T2w sequences; and the overall image quality for each complete examination. The evaluation of anatomic structures included the menisci, ligaments, and cartilage. A 5-point Likert scale was used with the following grading: 1 = not diagnostic; 2 = reduced diagnostic value; 3 = diagnostic; 4 = good; and 5 = excellent.

Pathological meniscal, ligamentous, and cartilage abnormalities were assessed by both readers. The location of meniscal tears or signal alterations (anterior/posterior horn or body of the medial/lateral meniscus) and the grade (normal/intra-substance signal abnormality = 0 and tear = 1) were determined on 2D coronal and sagittal PDw fs images, 2D sagittal T2w images, and 2D T1w images. Collateral and cruciate ligaments were scored as 0 if unremarkable or with minor degenerative changes and 1 if ligament tears were detectable. Cartilage lesions were classified using a modified Noyes grading of chondromalacia [20]. Four grades were differentiated: the signal heterogeneity and/or swelling (I°); less than a 50% defect (II°); more than a 50% defect (III°); and a full-thickness defect (IV°). If more than one defect was present, only the largest cartilage lesion was assessed.

### 2.3. Measurement of Contrast Ratios (CRs)

A quantitative analysis was performed calculating the contrast ratios (CRs) between the bone marrow and menisci, respectively, the cartilage, and between the cartilage and menisci. Measurements were only made on coronal PDw fs TSE sequence images with a placement of an 8 mm^2^ region of interest (ROI) in the medial condyle, medial meniscus, and in the cartilage of the medial condyle. If the structures were altered due to pathological conditions, the placement of the ROI was on the lateral side. The signal intensity (SI) for each structure obtained from coronal 2D PDw fs TSE images was used to calculate the CRs according to the formula CR = (SI _tissueA_ − SI _tissueB_)/(SI _tissueA_ + SI _tissueB_).

### 2.4. Statistical Analysis

All statistical tests were performed using R (version 4.2.1; The R Foundation for Statistical Computing, Vienna, Austria). The data of the qualitative image analysis were shown as the mean and median values with the interquartile range (IQR). To check the normal distribution of the continuous variable data, the Shapiro–Wilk test was used. The image quality and CRs were compared using the Wilcoxon rank sum test. Values of *p* < 0.05 were considered to indicate a statistical significance. To evaluate the intra- and inter-observer agreement, we performed a Cohen’s weighted kappa analysis (quadratic weights); agreement was categorized according to the Cohen’s kappa values as follows: 0–0.20 (poor); 0.21–0.40 (fair); 0.41–0.60 (moderate); 0.61–0.80 (substantial); and 0.81–1.00 (almost perfect) [21]. 

## 3. Results

### 3.1. Image Quality

Due to the reduced inter-reader agreement, in the following section the results of both readers are given. A summary of all qualitative image analyses and Cohen’s kappa is provided in Table 2.

In terms of PDw fs sequences, the parameter noise, edge sharpness, and contrast between the articular fluid and cartilage obtained a superior rating at 1.5T compared with 0.55T (*p* < 0.05). The mean values of the Likert scale at 0.55T were rated (reader 1/reader 2) 2.7/3.3 for noise, 3.7/3.9 for edge sharpness, and 4.1/4.4 for contrast between the articular fluid and cartilage. At 1.5T, the noise was rated 3.9/4.1, the edge sharpness was 4.1/4.3, and the contrast between the articular fluid and cartilage was 4.6/4.8. 

The overall image quality of the PDw fs and T2w sequences was rated significantly higher at 1.5T (4.0/4.4 for PDw and 4.2/4.6 for T2w) compared with 0.55T (3.0/3.7 for PDw and 3.7/4.2 for T2w). In contrast, there was no significant difference seen regarding the overall image quality of the T1w sequences between both field strengths. The overall image quality of the T1w images was rated 4.0/4.5 at 0.55T and 4.0/4.6 at 1.5T (*p* reader 1/reader 2 = 1.00/0.19) 

The average rating of the overall image quality, including all sequences of each examination, was 4.1/4.5 for 1.5T and 3.4/4.0 for 0.55T. The difference in the rating of the overall image quality was statistically significant (*p* < 0.05). Figure 1 shows an example examination of a healthy volunteer at 0.55T and 1.5T.

### 3.2. Visibility of Anatomical Structures and Detected Pathologies

The assessment of the anatomical structures was performed using PDw fs sequences with regard to the menisci, ligaments, and cartilage. At 0.55T, all anatomical structures were rated to be sufficiently assessable for diagnostic purposes. The mean values of the Likert scale at 0.55T were rated (reader 1/reader 2) 3.4/4.2 for menisci, 3.6/4.0 for ligaments, and 3.0/4.0 for cartilage. However, the visibility of all three anatomical structures was rated significantly better at 1.5T (*p* < 0.05). The results are given in Table 2.

Although the study participants had no acute knee pain, we detected several joint-related pathologies. 

With regard to the pathological ligamentous findings, we found one volunteer to have a reconstruction of the anterior cruciate ligament. Another volunteer showed severe bilateral mucoid degeneration of the anterior cruciate ligament (shown in Figure 2). No acute ligamentous tears were detected in the study population, either at 0.55T or at 1.5T.

From a total of 40 knee examinations, 12 chronic meniscal tears were diagnosed by both radiologists (see Figure 3). Each reader assessed 10 tears, 8 of them in accordance. Reader 1 detected 9 lesions at both field strengths, but only 1 at 0.55T. Reader 2 diagnosed 7 of the detected 10 meniscal tears at both field strengths; 2 only at 0.55T and 1 only at 1.5T. The resulting intra-reader agreement at 0.55T and 1.5T was calculated as kappa = 0.79 for reader 1 and kappa = 0.48 for reader 2. The inter-reader agreement at 0.55T was substantial (kappa = 0.66) and substantial to almost perfect at 1.5T (kappa = 0.78). The inter-reader agreement was substantial for 0.55T to almost perfect for 1.5T. The kappa values are given in Table 3.

In the study population, we diagnosed 28 cartilage lesions overall (grade I, *n* = 7; grade II, *n* = 10; grade III, *n* = 4; grade IV, *n* = 7; see Figure 4). A total of 23 lesions were detected by reader 1 and 26 by reader 2; among them, 23 were in accordance. Reader 1 assessed 20 cartilage lesions at both field strengths, 18 of them with the same graduation and only 3 at 0.55T (grade I°). Reader 2 diagnosed 15 of 26 lesions at both field strengths (8 with the same graduation). From the remaining 11 lesions, 5 were only detected at 0.55T and 6 only at 1.5T. The resulting intra- and inter-reader agreement was substantial to almost perfect, with kappa values between 0.6 and 0.89 (Table 3).

### 3.3. Measurement of Contrast Ratios

Comparative CR measurements of the cartilage and meniscus, bone marrow and meniscus, and cartilage and bone marrow showed no significant difference between 1.5T and 0.55T (*p* > 0.05), as shown in Table 4. 

## 4. Discussion

The aim of the present study was to subjectively and objectively compare the image quality of a standard knee MRI protocol from a new 0.55T MRI scanner with deep learning image reconstruction and a conventional 1.5T MRI scanner without this image reconstruction algorithm in a clinically reasonable and comparable acquisition time. The image quality at 0.55T yielded a comparable diagnostic quality, as rated by two blinded radiologists. However, the subjective image noise perception was rated higher by both readers. Except for the T1w images, the noise, edge sharpness, and contrast were significantly higher at 1.5T for the PDw fs TSE images and the overall image quality for the T2w TSE and PDw fs TSE images. The assessed contrast ratios were almost equal at both field strengths. With regard to the detected meniscal and cartilage defects, there was a high concordance between 0.55T and 1.5T.

To our knowledge, there are no current studies available comparing low-field MRI using deep learning algorithms and high-field MRI for knee imaging. Due to new developments in hardware and software, a comparison with older published studies has to be carefully undertaken [4]. Ghazinoor and colleagues reported in a review article in 2007 that the large majority of studies comparing knee examinations between low- and high-field scanners found no clinically significant differences in the detection of meniscal tears and tears of the anterior cruciate ligament [22]. In a more recent study, Leigheb and colleagues [15] demonstrated that low-field MRI at 0.3T with dedicated joint equipment yielded a diagnostic reliability for lesions of the menisci and cruciate ligaments. Our data concerning meniscal pathologies were in line with these studies, achieving a comparable and substantial inter-reader agreement. However, Leigheb et al. described a low diagnostic accuracy associated with a low inter-observer concordance in detecting cartilage lesions with their scanner, especially superficial grade I°–II° chondral injuries. There are only a few studies evaluating the articular surface by low-field MRI [15,22]. However, most studies documented an inferiority of low-field versus high-field scanners in detecting and grading cartilage lesions of the knee [16,23,24]. Ghazinoor and colleagues concluded that cartilage abnormalities are more difficult to evaluate on low-field MRI scanners [22]. In contrast, both readers in our study reached a substantial to almost perfect intra- and inter-reader agreement for cartilage lesions. This finding indicates a high potential of modern low-field MRI with deep learning algorithms for the detection of cartilage pathologies. Nonetheless, further clinical studies compared with arthroscopy findings are mandatory to further prove these initial results.

With respect to magnetic field strength, several studies showed an improvement in the subjective assessment of the image quality at 3T versus 1.5T [6,25]. In contrast, there are no recent studies comparing low-field MRI with standard MRI at > 1T. In our study, all evaluated items of the image quality at 0.55T were considered to be diagnostic, but inferior to 1.5T, except for the T1w images. This was in line with the physical expectation because the SNR increases with a higher field strength, resulting in a better visualization of anatomical structures and image quality [26]. However, deep learning algorithms can increase the SNR to produce a high image quality to preserve the image quality whilst reducing the acquisition time [27,28]. These deep learning reconstruction methods use neural networks to learn robust transformation mappings from the sensor space to the image domain [29]. A possible explanation for the above-mentioned comparable evaluation of the T1w TSE sequences at 0.55T versus 1.5T may be because the T1 value is shorter at lower magnetic field strengths combined with an increased T1 contrast. The T1 relaxation values in various tissues are larger at low-field strengths, which is advantageous for T1 tissue contrast [7,30]. The T1 difference of different tissues, e.g., fat and muscle tissue, increases at lower field strengths [31]. An optimal T1 contrast was described at a field strength of 0.23T and a frequency of 10 MHz by Fischer et al. [32]. Although all assessed contrast ratios, as a parameter of objective image quality, showed no significant difference between both field strengths, a high subjective image noise perception at 0.55T was especially obvious. The absolute scores of the image quality assessment of the two readers differed within a normal range of variation in the diagnostic range because there are intra- and inter-individual variations when using a 5-point scale. It should be noted that regardless of the field strength, all knee examinations were of a diagnostic to excellent image quality. In this context, we discussed only the poor to fair inter-reader agreement for image quality in our study. It is important to note that on average, reader 1 rated the items half to a full point lower than reader 2. Therefore, both supported the main finding, but differed in the absolute graduation. 

Our findings should be interpreted within the context of the study’s limitations. First, although both readers were blinded to the field strength, the characteristic appearance of sequences at 0.55T with a higher image noise perception allowed the used field strength to be recognized. An additional limitation was the use of the manufacturer’s predefined standard protocols, which could be optimized even further. However, the goal of the study was to compare the performance of 0.55T and 1.5T in a clinical setting without using customized protocols. Finally, the acquired pathological findings of the menisci and cartilage were limited due to the study population and the study design. These findings were derived from “healthy” volunteers and not correlated with arthroscopy as a gold standard. Due to a lack of pathological findings, no evaluation of the ligaments could be made. Consequently, further studies are required to substantiate these initial findings. Although we did not compare our 0.55T images with 1.5T images enhanced with an AI-based reconstruction, it is not expected that the image quality from a 0.55T system enhanced with an AI reconstruction will ever be comparable with higher field systems also employing AI reconstruction technology.

In conclusion, this study indicated that deep learning-reconstructed TSE imaging at 0.55T yielded a diagnostic image quality for knee MRI compared with standard 1.5T MRI. The overall image quality was rated significantly better at 1.5T due to the increased subjective perception of image noise at 0.55T. The detection of cartilage and meniscal disorders was comparable between 0.55T and 1.5T without an overall loss of diagnostic information. Finally, we found no diagnostic limitations of 0.55T MRI enhanced by deep learning algorithms for the examination of knee joints compared with conventional 1.5T MRI. Nevertheless, image reconstruction algorithms cannot fully compensate for the physically lower signal-to-noise ratio of MRI at 0.55T compared with 1.5T; thus, the user must accept visible image noise or longer acquisition times.

## Figures and Tables

**Figure 1 jcm-12-01916-f001:**
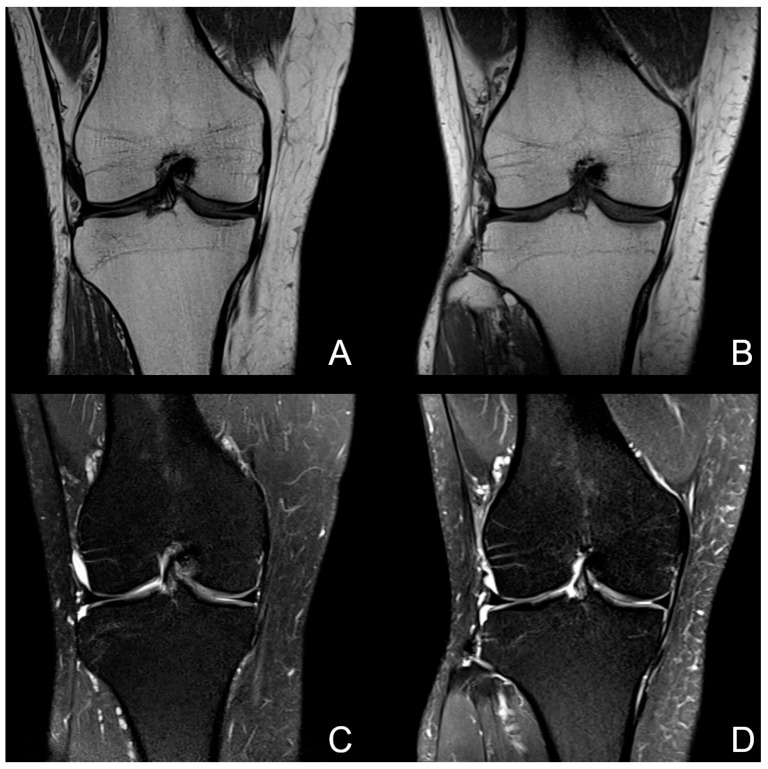
Image example of a knee examination of a healthy volunteer at 0.55T on the left (**A**,**C**) and 1.5T on the right (**B**,**D**). Coronal T1w sequences (upper row) show a comparable image quality with homogenous bone marrow and clearly definable meniscal and cartilage structures. The lower row images are examples of a coronal PDw fs sequence. The extent of noise in this case was comparable between both field strengths, but the signal intensity of small vascular structures within the soft tissue and bone marrow was lower in the 0.55T image (**C**). Note the good delineation of cartilage and menisci at both field strengths without a decrease in the diagnostic value.

**Figure 2 jcm-12-01916-f002:**
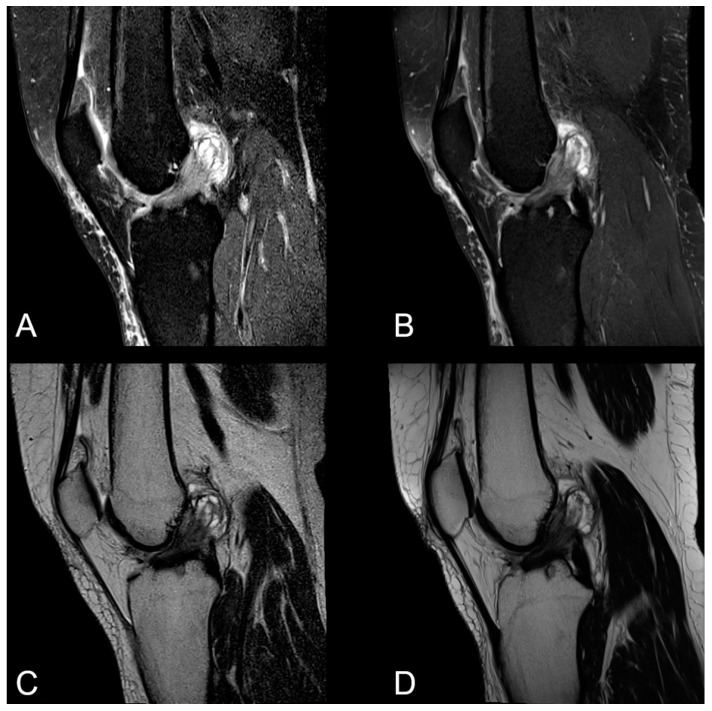
Severe mucoid degeneration of the anterior cruciate ligament at 0.55T on the left (**A**,**C**) and 1.5T on the right (**B**,**D**). Sagittal PDw fs (upper row) and sagittal T2w sequences (lower row) are given. The pathological widening of the ligament, combined with a signal increase and associated ganglions, is clearly detectable at both field strengths. However, the image noise is enhanced in both sequences at 0.55T compared with 1.5T.

**Figure 3 jcm-12-01916-f003:**
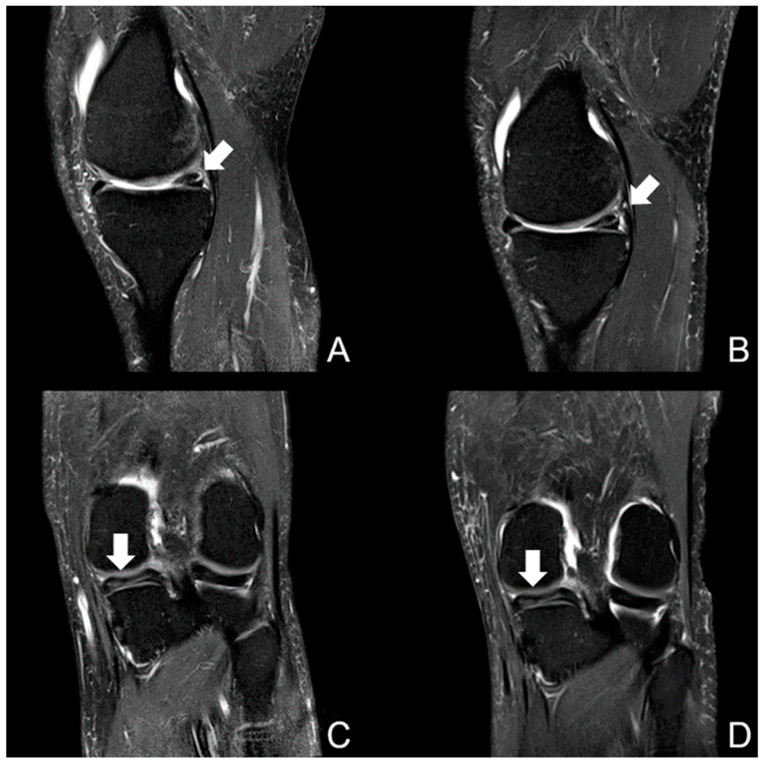
Image example of a sagittal PDw fs (upper row) and a coronal PDw fs sequence (lower row) at 0.55T on the left (**A**,**C**) and 1.5T on the right (**B**,**D**). This was an examination of a 72-year-old male participant, showing a horizontal tear in the posterior horn of the medial meniscus (white arrows). In sagittal orientation, the slices were slightly different; however, they clearly demonstrated the meniscal tear. Despite a moderate enhanced noise, there was no diagnostic loss at 0.55T.

**Figure 4 jcm-12-01916-f004:**
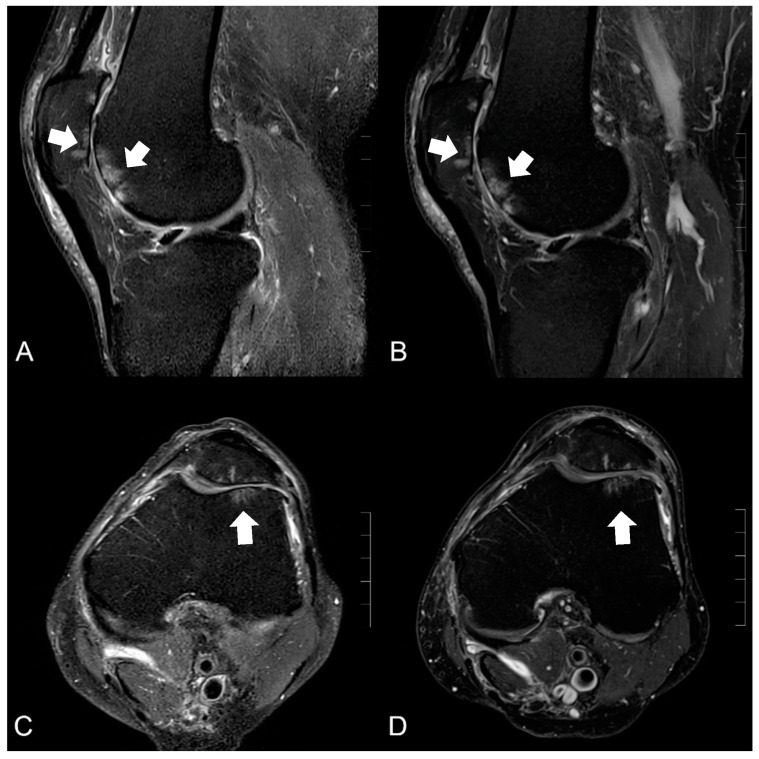
Image example of a knee MRI at 0.55T (left, **A**,**C**) and 1.5T (right, **B**,**D**) in PDw fs sagittal and axial orientation of a 68-year-old man with osteoarthritis of the femoropatellar joint. The cartilage defects (classified as grade IV°) with adjacent bone edema (white arrows) were detectable in both field strengths whereas the edge sharpness of the residual cartilage on the medial side was slightly reduced at 0.55T compared with 1.5T.

**Table 1 jcm-12-01916-t001:** MRI acquisition parameters.

	Coronal PDw fs		Transversal PDw fs		Sagittal PDw fs		Coronal T1w		Sagittal T2w	
Imaging Parameter	0.55T	1.5T	0.55T	1.5T	0.55T	1.5T	0.55T	1.5T	0.55T	1.5T
TR repetition time (msec)	2950	3500	3220	2700	2950	3420	450	624	3740	5660
TE echo time (msec)	45	40	44	37	45	41	13	11	108	86
Slice thickness (mm)	3	3	3	3	3	3	3	3	3	3
Slice spacing (%)	10	10	10	10	10	10	10	10	10	10
Field of view (cm)	17	17	17	17	17	17	17	17	17	16
No. of slices	30	30	32	35	30	30	30	30	30	30
Reconstructed pixel spacing (mm^2^)	0.09	0.25	0.09	0.16	0.09	0.25	0.09	0.25	0.09	0.16
Voxel size (mm^3^)	0.3 × 0.3 × 3	0.5 × 0.5 × 3	0.3 × 0.3 × 3	0.4 × 0.4 × 3	0.3 × 0.3 × 3	0.5 × 0.5 × 3	0.3 × 0.3 × 3	0.5 × 0.5 × 3	0.3 × 0.3 × 3	0.4 × 0.4 × 3
Acceleration	GRAPPA + SMS	GRAPPA	GRAPPA + SMS	GRAPPA	GRAPPA + SMS	GRAPPA	GRAPPA + SMS	GRAPPA	GRAPPA + SMS	GRAPPA
Acceleration factor	4	2	4	2	4	2	4	2	4	2
Bandwidth (Hz/pixel)	100	150	100	181	100	150	126	150	126	149
Time of acquisition (min)	04:00	02:04	03:11	03:32	04:00	02:08	04:22	03:45	04:05	04:05
Turbo factor	9	7	9	7	9	7	3	1	17	15
Deep resolve	Yes	No	Yes	No	Yes	No	Yes	No	Yes	No

**Table 2 jcm-12-01916-t002:** Image quality and inter-reader agreement at 0.55T and 1.5T. The results are shown as mean [median (interquartile range)]. Cohen’s kappa gave an inter-reader agreement between both readers.

Sequence	Item	Reader 1			Reader 2			Cohen’s Kappa	
		0.55T	1.5T	*p*-Value	0.55T	1.5T	*p*-Value	0.55T	1.5T
PDw fs TSE	Noise	2.70 [3 (2–3)]	3.88 [4 (4–4)]	<0.001	3.28 [3 (3–4)]	4.05 [4 (4–4)]	< 0.001	0.06	0.40
	Edge sharpness	3.70 [4 (3–4)]	4.13 [4 (4–4)]	<0.001	3.90 [4 (3–4)]	4.33 [4 (4–5)]	0.006	0.05	−0.14
	Contrast fluid/cartilage	4.05 [4 (4–4)]	4.63 [5 (4–5)]	<0.001	4.40 [4 (4–5)]	4.75 [5 (5–5)]	0.003	0.02	0.30
	Menisci	3.43 [3 (3–4)]	4.25 [4 (4–5)]	<0.001	4.15 [4 (4–5)]	4.65 [5 (4–5)]	0.004	0.12	0.24
	Ligaments	3.55 [4 (3–4)]	4.13 [4 (4–4)]	<0.001	4.00 [4 (4–4)]	4.58 [5 (4–5)]	< 0.001	0.28	0.36
	Cartilage	3.00 [3 (2–4)]	4.18 [4 (4–4.25)]	<0.001	3.98 [4 (3–5)]	4.55 [5 (4–5)]	0.002	0.27	0.39
	Overall image quality	3.03 [3 (3–3)]	3.98 [4 (4–4)]	<0.001	3.73 [4 (3–4)]	4.35 [4 (4–5)]	< 0.001	0.29	0.29
T1w TSE	Overall image quality	3.95 [4 (4–4)]	3.95 [4 (4–4)]	1.00	4.45 [4 (4–5)]	4.60 [5 (4–5)]	0.19	0.08	−0.06
T2w TSE	Overall image quality	3.73 [4 (3–4)]	4.23 [4 (4–5)]	<0.001	4.18 [4 (4–5)]	4.63 [5 (4–5)]	0.006	0.16	0.26
Overall image quality		3.38 [3 (3–4)]	4.08 [4 (4–4)]	<0.001	4.03 [4 (4–4)]	4.48 [4.5 (4–4.5)]	0.004	0.27	0.15

**Table 3 jcm-12-01916-t003:** Intra- and inter-reader agreement of meniscal and cartilage pathologies at 0.55T and 1.5T.

Item	Cohen’s Kappa		Cohen’s Kappa	
	Reader 1	Reader 2	0.55T	1.5T
Meniscus tear	0.79	0.48	0.66	0.78
Cartilage defects	0.8	0.6	0.7	0.89

**Table 4 jcm-12-01916-t004:** Contrast ratios of cartilage, menisci, and bone marrow at 0.55T and 1.5T. Q1: first quartile; Q3: third quartile.

Anatomic Structure	Field Strength	Mean	Median	Q1	Q3	*p*-Value
Cartilage/meniscus	0.55	0.84	0.86	0.8	0.89	0.97
	1.5	0.83	0.86	0.79	0.9	
Bone/meniscus	0.55	0.56	0.6	0.47	0.68	0.92
	1.5	0.56	0.59	0.43	0.72	
Cartilage/bone marrow	0.55	0.53	0.53	0.48	0.6	0.89
	1.5	0.53	0.54	0.46	0.59	

## Data Availability

The data are available from the corresponding author on reasonable request.

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
