# Peer review of "Diagnostic Image Quality of a Low-Field (0.55T) Knee MRI Protocol Using Deep Learning Image Reconstruction Compared with a Standard (1.5T) Knee MRI Protocol"

_jcm, 2023, doi:10.3390/jcm12051916_

Round 1

Reviewer 1 Report

Authors present manuscript on clinically very interesting topic: application of “low field MRI”. Authors listed several arguments why is application of low field so important, particularly when combined with AI.

Study design is correct and reasonably compares images of the human knee acquired at low field 0.55T scanner with similar images acquired at 1.5T MRI.

Group of volunteers without acute knee pain or acute injury was examined. Subjective evaluation by two radiologists was performed and quantitative comparison was performed by CR evaluation.

List of references is reasonable covering relevant publications.

I have following comments where I expect explanation:

1.  It is well known, that T1 is slightly growing with the magnetic field strength. Therefore the TR at 1.5T should always be higher compared to TR at 0.55T. According to the Table 1, it is true for Coronal PDW fs, Sagittal PDW fs, Coronal T1w and Sagittal T2w. In case of Transversal PDW fs it is not true. Could you explain it, please ?

2.  You compared images 0.55T with AI to the 1.5T without AI. It would be more fair to compare 0.55T with AI to the 1.5T with AI. Could you comment on it, please ?

3.   You performed subjective and also objective (quantitative) evaluation based on the examination of non-symptomatic volunteers. It would be very interesting to perform similar study on the patient, having focal cartilage defect. Do you plan to perform such study in future ?

Author Response

Manuscript ID: jcm-2203626

Dear reviewers of the Journal of Clinical Medicine,

Thank You for reviewing and improving our manuscript „Diagnostic image quality of a low-field (0.55T) knee MRI protocol using deep-learning image reconstruction compared to a 3 standard (1.5T) knee MRI protocol“ and for the opportunity to perform minor revisions.

Please let us add another co-author (Mr. Frohwein, Siemens Healthineers) because he supported our work with technical support and collaborated with Iram Shazadi for the statistical data analysis.

According to your reviewer comments we improved/changed the manuscript as answered in the comments beside the main manuscript and corrected it as suggested.

Find attached our answers to the additional reviewer questions:

„It is well known, that T1 is slightly growing with the magnetic field strength. Therefore the TR at 1.5T should always be higher compared to TR at 0.55T. According to the Table 1, it is true for Coronal PDW fs, Sagittal PDW fs, Coronal T1w and Sagittal T2w. In case of Transversal PDW fs it is not true. Could you explain it, please?“

We used MR sequenced provided by the manufacturer and did not check or adapt the TR of the 2D PD fs TSE sequence. Of course we should have used a TR of 2950 msec as done in the coronal and sagittal scans. There is no specific reason fort hat. Transversal images were mainly used for the assessment of the patellar cartilage. CR measurements derive from the coronal PD fs series.

  1. You compared images 0.55T with AI to the 1.5T without AI. It would be more fair to compare 0.55T with AI to the 1.5T with AI. Could you comment on it, please?

We wanted to compare 0.55T with AI (as intended) to a standard 1.5T MRI without AI, because these systems are now widely used in hospitals and radiological practices. We wanted to give the radiological colleages a fair impression of what can be expected from a low field MR system, that would not be competitive to 1.5T without its AI features.

Further studies are needed which compare MR images from the same scan with and without AI applications to recognize probable pitfalls and artifacts. We intended to do a volunteer study with sequences and MR scanners right from the shelf (because both were newly installed in our institution). Both scanners differ in field strength, coil design, bore diameter, sequence parameters are other factors. To some extent a really fair comparison of imaging and MR physics of both machines remains impossible.

  1. You performed subjective and also objective (quantitative) evaluation based on the examination of non-symptomatic volunteers. It would be very interesting to perform similar study on the patient, having focal cartilage defect. Do you plan to perform such study in future?

Indeed this study is a small prospective pilot study. A comparison study including patients with known cartilage defects would be beneficial for radiologist with muscolosceletal focus. Right now we perform comparative imaging studies of patients with lung diseases, strokes and with metal implants oft he spine, because other groups have already pointed out some benefits of the lower field strength empowered by deep learning algorithms for better image quality.

  1. It's an interesting article about the quality of images and importance of diagnosis. For meniscal and cartilage problems, we don't necessary need an high quality of examen. This will help prescriptor of MRI, to choose an old machine if available, to do the same job.

Thank You for putting the intention of our manuscript into a nutshell. The new 0.55T MR whole body scanner is lightweight, contains only 0,7l helium, needs only 24 qm of space and has of course some drawbacks for detailed imaging of fine anatomical structures in a clinically reasonable scan time. Although we have a scientific cooperation with the manufacturer and two co-authors are employees of the manufacturer, this study was driven by active radiologist interested in the question, what image quality can be provided by 0.55T MRI supported by AI.

Yours sincerely,

Christoph Moenninghoff, M.D.

senior author and radiologist

Author Response

(The authors gave the same response as above.)

Reviewer 3 Report

It's an interesting article about the quality of images and importance of diagnosis. For meniscal and cartilage problems, we don't necessary need an high quality of examen. This wil help prescriptor of MRI, to choose an old machine if available, to do the same job.

Author Response

(The authors gave the same response as above.)
